# ROAMM: A Benchmark Dataset for Multimodal Human Attention Decoding and EEG-to-Text Modeling During Naturalistic Reading

**Haorui Sun** [1]  **Ardyn Olszko** [1]  **Niharika Singh** [1]  **David Jangraw** [1]

## Abstract

We present Reading Observed At Mindless Moments (ROAMM), a multimodal dataset comprising 50 hours of simultaneous EEG and eye-tracking recordings collected during naturalistic multi-page reading from 44 participants. ROAMM includes synchronized physiological recordings, eye-movement events, page-level comprehension scores, and span-level mind-wandering (MW) annotations obtained using a retrospective self-report paradigm. We introduce a standardized leave-one-subject-out benchmark for MW detection and achieve up to 0.609 AU-ROC using supervised models. We additionally evaluate EEG-to-text decoding on reading segments with and without MW labels, showing that decoding performance decreases during MW episodes. ROAMM enables research on MW detection, EEG-to-text decoding, multimodal representation learning, and attention-related degradation of language representations during naturalistic reading.

## 1. Introduction

Advances in noninvasive recording techniques such as eye-tracking (ET) and electroencephalography (EEG) have enabled simultaneous measurement of gaze behavior and brain activity with high temporal resolution and made it feasible to study cognition in naturalistic task settings (Zhu & Lv, 2023). This multimodal setup provides a direct way to study how visual attention and neural dynamics jointly support perception and language processing, and has motivated growing interest in decoding task-relevant information from neural and behavioral signals (Mathis et al., 2024). In visual search and image-viewing paradigms, eye tracking provides precise measurements of gaze trajectories and fixation locations, enabling models to predict gaze behavior from brain activity and characterize neural correlates of visual attention (Modesitt et al., 2023; Pšurný et al., 2024; Moreno-Alcayde et al., 2025). In reading, eye tracking is particularly valuable because fixations can be mapped to individual words, enabling fine-grained alignment between linguistic input and neural responses (Dimigen et al., 2011; Hollenstein et al., 2018; 2020) and supporting the development of neural language decoding models (Duan et al., 2023; Wang & Ji, 2022; Zhou et al., 2024; Liu et al., 2024; Wang et al., 2024). Such multimodal decoding approaches have potential applications in both cognitive neuroscience and future brain–computer interface (BCI) systems for communication.

A key challenge for decoding brain activity in realistic settings is that the internal cognitive state varies substantially over time. Attention is not stable: our minds frequently drift away from the task at hand. Mind-wandering (MW), often defined as task-unrelated thought, has been reported to occupy 30–60% of daily life (Killingsworth & Gilbert, 2010) and is associated with degraded task performance (van Vugt & Broers, 2016) and comprehension (Smallwood et al., 2008; Unsworth & McMillan, 2013). We hypothesize that during reading, MW can disrupt linguistic processing even when outward reading behavior appears normal (Reichle et al., 2010), thereby confounding both behavioral analyses and neural decoding models. This motivates the need for datasets that capture reading under realistic conditions while providing explicit attention-state annotations to support attention-aware modeling and evaluation (Ho & Griffiths, 2022).

To address this challenge, we present ROAMM, a large-scale dataset of simultaneous EEG and ET collected during naturalistic multi-page reading (Figure 1). To our knowledge, ROAMM is the first publicly available dataset that combines co-registered EEG and ET with span-level MW onset/offset labels in a naturalistic reading paradigm. To demonstrate ROAMM's utility as a reference dataset, we evaluated its performance for MW detection using established EEG and eye-tracking feature pipelines and supervised hybrid classifiers under leave-one-subject-out (LOSO) evaluation. We further apply an existing EEG-to-text decoding approach

[1]Department of Electrical and Biomedical Engineering, University of Vermont, Burlington, VT, USA. Correspondence to: Haorui Sun <haorui.sun@uvm.edu>.

*Proceedings of the 43rd International Conference on Machine Learning*, Seoul, South Korea. PMLR 306, 2026. Copyright 2026 by the author(s).

(Wang & Ji, 2022) to evaluate models trained with and without MW-labeled segments. We show that decoding performance drops when MW-labeled segments are included in both training and testing, highlighting the importance of accounting for attention fluctuations in neural language decoding. Overall, ROAMM provides a benchmark dataset and evaluation protocol for studying attention variability and neural decoding in realistic reading. Our findings not only offer insight into the cognitive signatures of MW but also support the development of more robust attention-aware decoding models for real-world BCI applications.

Our contributions are as follows:

- **Dataset:** 50 hours of simultaneous EEG and ET recorded during naturalistic multi-page reading from 44 participants, with annotations including eye events, page-level comprehension scores, and span-level MW labels obtained via a retrospective self-report paradigm. https://openneuro.org/datasets/ds007629

- **MW detection:** Hybrid EEG and ET feature pipelines with supervised classifiers evaluated under LOSO, showing generalization across individuals and achieving up to 0.609 AUROC.

- **EEG-to-text decoding demo:** Demonstration that decoding performance drops on MW-labeled segments, motivating attention-aware modeling for real-world BCI applications. https://github.com/GlassBrainLab/roamm_ml

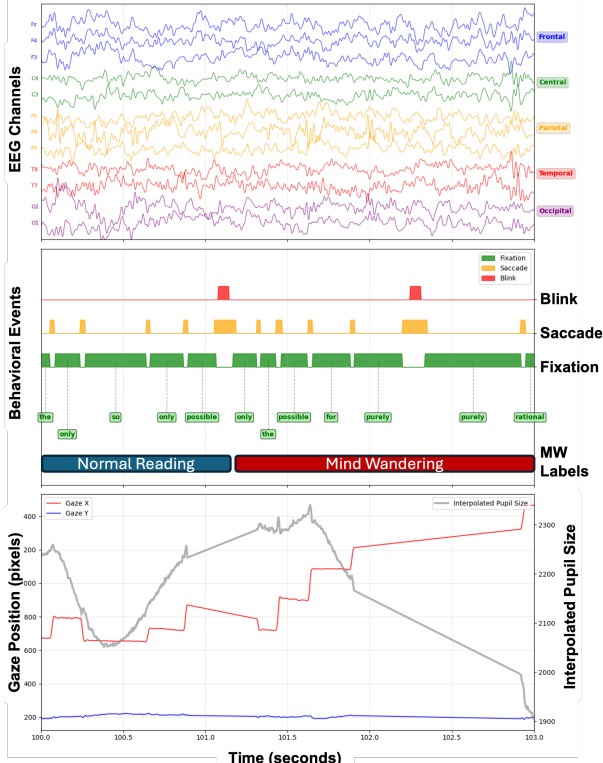

*Figure 1.* Example of 3 s of simultaneous preprocessed EEG and eye-tracking data from ROAMM. EEG signals from representative channels across different brain regions are shown. ET signals, including gaze position and pupil size, are plotted alongside annotated eye events, the corresponding fixated words and span-level MW labels.

## 2. Related Work

Because ROAMM is a multimodal dataset, it can be applied to a wide range of research questions across multiple fields. In this paper, we focus on two main themes: MW detection and simultaneous EEG and ET datasets for neural decoding, and organize related work accordingly.

### 2.1. Mind-Wandering Detection

Mind-wandering detection has attracted growing attention because MW occurs frequently in everyday tasks and is linked to reduced task performance (Smallwood et al., 2008; Unsworth & McMillan, 2013). However, capturing MW is inherently challenging due to its spontaneous and internally generated nature. Most prior work relies on self-reports (Chu et al., 2023) or thought probes (Greve & Was, 2022) to obtain ground-truth labels, but these methods offer only coarse temporal information and typically indicate that MW occurred within an imprecise time window before the report, for example, by using fixed-length windows such as 2 or 5s preceding probes (Reichle et al., 2010). Within these

paradigms, some differences in behavior and neural activity have been observed during MW, including changes in gaze patterns (Reichle et al., 2010) and EEG markers such as event-related potentials (ERPs) and bandpower (Kam et al., 2022). These signatures have been widely used as features in supervised models to distinguish MW from on-task states (Faber et al., 2017; Dong et al., 2021).

Despite these advances, precisely determining when mind-wandering occurs remains an open challenge. Fine-grained temporal localization of MW is essential for understanding the neural mechanisms underlying attention fluctuations and for refining theoretical models of human attention (Smallwood & Schooler, 2006; Smallwood, 2013). This level of understanding is important not only for cognitive neuroscience, but also for developing attention-aware machine learning systems that can generalize across tasks and real-world settings (Roda & Thomas, 2006). In this context, ROAMM makes a unique contribution by providing time-resolved MW onset and offset labels during a naturalistic reading task. Compared to commonly used sustained-attention paradigms such as breath focusing (Rodriguez-Larios & Alaerts, 2020)

or simple stimulus–response tasks (Jin et al., 2019), naturalistic reading is more engaging and more closely reflects everyday cognition. For the machine learning community, this enables realistic benchmarking of multimodal models under attention variability and supports the development of algorithms that explicitly model cognitive state rather than assuming uniformly attentive data.

## 2.2. Simultaneous EEG and Eye-Tracking Recordings

Several prior studies and datasets have leveraged simultaneous EEG and ET to capture gaze behavior under different experimental paradigms. For example, datasets such as EEGEyeNet (Kastrati et al., 2021) and consumer-grade (CG) EEGET (Afonso & Heinrichs, 2025) recorded eye movement direction, speed, and absolute gaze position alongside EEG signals, enabling the prediction of gaze behavior from neural activity and the characterization of task-dependent gaze patterns.

A key advantage of eye-tracking is that it not only captures continuous gaze dynamics, but also provides precise information about where participants are looking on the screen at each moment, effectively revealing the visual stimulus driving neural responses. For instance, a dataset, EEGET-RSOD tracks gaze and EEG while expert annotators search for targets in large-scale remote sensing images (He et al., 2025), supporting the development of machine learning models for target detection and visual search in complex imagery.

In reading research, fixation-to-word alignment provides an important structural link between neural activity and specific linguistic units. With recent advances in natural language processing (NLP) and large language models (LLMs), this alignment has become increasingly valuable for neural decoding studies. However, only a small number of public datasets provide simultaneous EEG and ET during reading, most notably ZuCo 1.0 (Hollenstein et al., 2018) and ZuCo 2.0 (Hollenstein et al., 2020) . These datasets focus on sentence-level reading and provide carefully preprocessed EEG and ET data, including fixation-to-word mappings and word-level EEG features. Due to their high data quality and clear experimental design, ZuCo datasets have become widely used in the field, advancing our understanding of neural and ocular processes during reading and enabling a growing body of EEG-to-text decoding models and architectures.

ROAMM builds on these strengths while extending the scope of existing resources. Like ZuCo, ROAMM provides high-quality, co-registered EEG and ET data with fixation-to-word alignment. In addition, ROAMM supports free-viewing, multi-page reading that preserves uninterrupted, naturalistic reading behavior. For the machine learning community, this distinction is important: ROAMM exposes models to the variability, noise, and attention fluctuations inherent in real-world cognition, rather than the tightly controlled conditions of sentence-level paradigms. Moreover, ROAMM uniquely provides span-level, time-resolved mind-wandering annotations, enabling attention-aware modeling and evaluation. This allows ML practitioners to investigate how attention variability influences multimodal representations, decoding robustness, and generalization, and to develop models that explicitly account for cognitive state rather than assuming uniformly attentive input.

*Table 1.* Comparison of ROAMM with related EEG and ET datasets. Recording length refers to valid EEG and ET data only and excludes preparation, instructions, and breaks. "–" indicates that recording duration was not directly reported in the original publication.

| Dataset | # Participants | Task | Length |
|---|---|---|---|
| EEGEyeNet | 356 | Symbol viewing | 47 h |
| CG-EEGET | 113 | Gaze movement | 12 h |
| EEGET-RSOD | 38 | Target search | – |
| ZuCo 1.0 | 12 | Sentence reading | – |
| ZuCo 2.0 | 18 | Sentence reading | – |
| **ROAMM** | **44** | **Page reading** | **50 h** |

## 3. ROAMM Dataset

### 3.1. Participants

We recruited 58 fluent English-speaking participants from the University of Vermont. All participants provided informed consent under a protocol approved by the university Institutional Review Board. Participants reported no family history of neurological disorders or epilepsy. ROAMM is fully de-identified, with participant-identifying information stored separately from the released dataset.

Fourteen participants were excluded due to incomplete experimental sessions, recording issues, monocular-only eye-tracking data, or missing demographic information, resulting in a final sample of 44 participants. Participant ages ranged from 18 to 64 years (mean = 22.6, SD = 7.8, median = 20, mode = 19). Demographic annotations additionally include gender, handedness, and self-reported ADHD status. Full demographic statistics are provided in the dataset repository and summarized in Appendix Table 5.

### 3.2. Reading Materials

The dataset was collected using the ReMind retrospective self-report paradigm introduced in prior work (Sun et al., 2026). We selected five Wikipedia articles (2015): *Pluto*, *Prisoner's Dilemma*, *Serena Williams*, *History of Film*, and *Voynich Manuscript*. Articles were chosen to be broadly understandable while minimizing dependence on prior topic knowledge.

To standardize presentation, articles were cleaned of images

and unnecessary formatting, then divided into 10 pages each (approximately 220 words per page; Table 2). Pages were rendered using a custom Python script as 16 lines of black Courier text on a gray background.

*Table 2.* Reading material statistics in the ROAMM dataset.

| STORY | # SENTENCES | # WORDS |
|---|---|---|
| PLUTO | 97 | 2125 |
| THE PRISONER'S DILEMMA | 101 | 2148 |
| SERENA WILLIAMS | 99 | 2269 |
| THE HISTORY OF FILM | 95 | 2141 |
| THE VOYNICH MANUSCRIPT | 95 | 2156 |

### 3.3. ReMind Paradigm

The experiment was implemented in PsychoPy (Peirce et al., 2019). Each participant completed five reading runs, one per article, presented in randomized order. Participants read at their own pace without page time limits and were not allowed to revisit previous pages to preserve first-pass reading behavior.

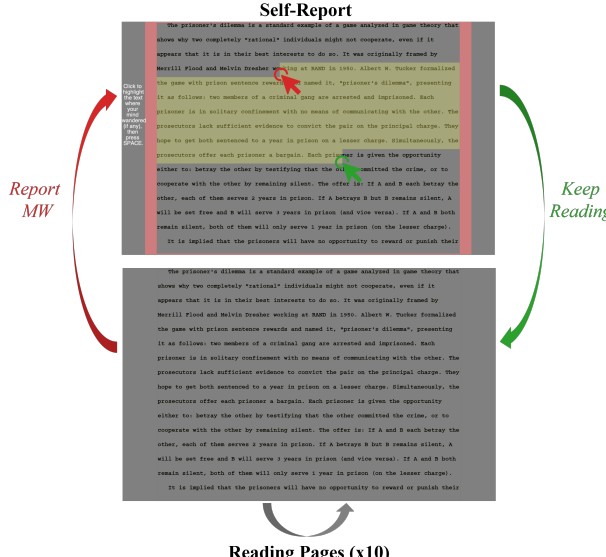

*Figure 2.* Naturalistic reading task with retrospective self-report paradigm.

Participants were instructed to press the "F" key whenever they became aware of mind-wandering (MW). After reporting MW, they were redirected to a dedicated annotation screen where they selected the words corresponding to the perceived onset and offset of the MW episode (Figure 2). If MW was believed to begin on the previous page, participants selected the first word on the current page. Following annotation, participants resumed reading from the same page. To simplify labeling, at most one MW report was permitted per page.

To encourage attentive reading, participants answered one multiple-choice comprehension question per page after completing each article. Additional details regarding the MW definition, the rationale for using self-reports rather than thought probes, and example comprehension questions are provided in the Appendix and Sun et al. (2026).

### 3.4. Data Acquisition

The experiment was conducted in a soundproof booth to minimize distractions (Figure 3). PsychoPy sent page-onset triggers to both EEG and ET systems to keep the data streams aligned.

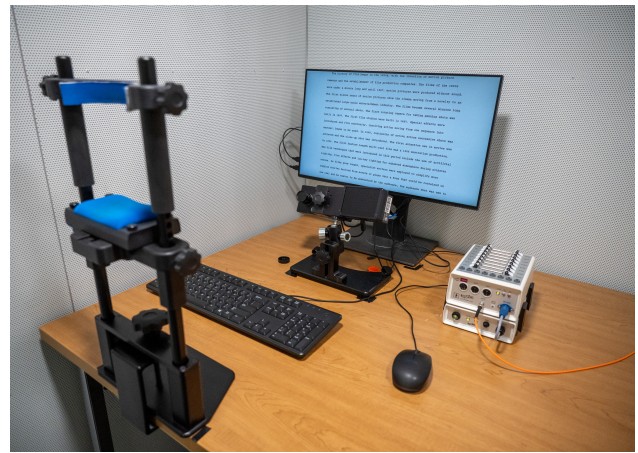

*Figure 3.* Recording setup.

**Eye-Tracking.** We used an SR Research EyeLink 1000 Plus eye tracker to record binocular eye movements and pupil area at 1000 Hz. Each run began with calibration, repeated until EyeLink reported good accuracy (worst error $< 1.5°$, average error $< 1.0°$).

**EEG.** We recorded simultaneous EEG using a BioSemi ActiveTwo 64-channel system at 2048 Hz. Before starting the task, we ensured all electrodes had stable connections by checking impedances and correcting any channels with unusually high values.

### 3.5. Preprocessing and Annotations

For eye tracking, we followed standard procedures: gaze position and pupil size samples were flagged during blinks (±100 ms around each blink), and rapid pupil artifacts were detected using a median absolute deviation (MAD) procedure (Geller et al., 2020). These samples were then linearly interpolated. EEG data were preprocessed in EEGLAB (Delorme & Makeig, 2004): signals were resampled to 256 Hz, re-referenced to a common average reference, band-

pass filtered (0.5–50 Hz), and bad channels were labeled using EEGLAB's automatic channel labeling procedure and interpolated. Ocular and muscle artifacts were removed using independent component analysis (ICA) with standard EEGLAB parameters (Delorme et al., 2007).

EEG and ET streams were synchronized using page-onset markers. Eye-tracking data were aligned to the EEG time base by downsampling via the real-time arrays, and eye events (fixations[1], saccades[2], and blinks) were mapped onto the EEG timeline using their start and end timestamps. We defined first-pass reading as the initial continuous period during which participants read each page (see Figure 4). For pages without a MW report, the entire page was labeled as first-pass reading. For pages with a MW report, first-pass reading was defined as the interval from page onset to the MW report time.

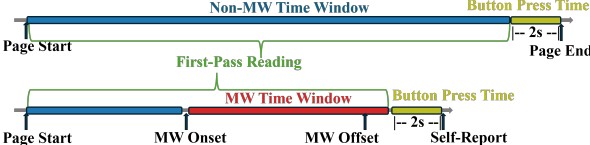

*Figure 4.* Time schematic of non-MW and MW reading pages, along with the corresponding timing annotations used in ROAMM.

To reduce contamination from motor responses and page transitions, we excluded the final 2 s preceding each button press (either an MW report or a page advance), consistent with prior work (Faber et al., 2017). This exclusion mitigates report-related artifacts, including vertical eye movements and motor-preparatory activity associated with impending button presses. Including these intervals could artificially inflate classification performance by capturing motor-related signals rather than MW-related cognitive states. By removing these periods, we ensure that the benchmark more accurately reflects physiological markers associated with MW.

Fixations were mapped to individual words using their screen coordinates (Figure 5). Fixations falling outside the bounds of any word were assigned to the word whose bounding box was closest to the fixation if those bounds were < 100 pixels away. MW onset was inferred as the start time of the first fixation on the self-reported MW onset word. Because MW offset times can be inconsistent when defined using self-reported offset words versus the report timestamp, and to reduce artifacts associated with reporting movements, we defined MW offset as 2 s before the MW report button press. All samples within the resulting

MW interval were labeled as MW. Using this procedure, participants reported 1,001 MW episodes across the dataset, corresponding to approximately 45.5% of 2,200 reading pages. The median MW duration was 6.14 s, and the mean duration was 7.99 s. Additional analyses of MW frequency and duration are reported in Sun et al. (2026).

The full dataset scale is summarized in Appendix Table 6. Detailed descriptions of the dataset structure and column definitions are provided in Appendix Table 7.

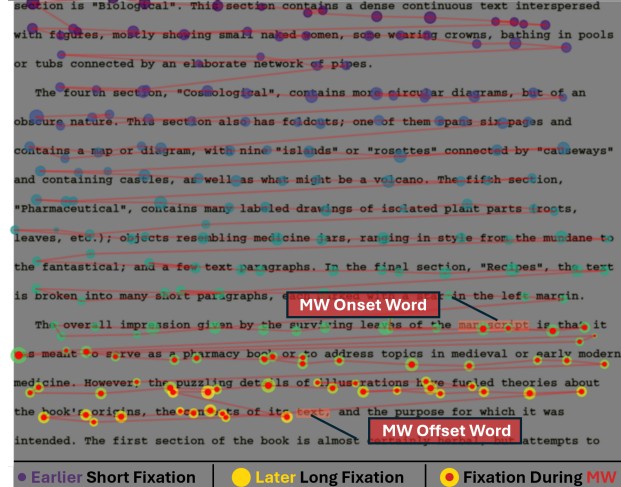

*Figure 5.* Eye-tracking data from a reading page with a reported MW episode. Self-reported MW onset and offset words are highlighted in red. Fixations are shown as dots, where circle size indicates fixation duration and color indicates chronological fixation order. Fixations within the MW interval are colored red. Red lines connecting fixation points represent saccades.

### 3.6. Data Validation

To validate the temporal synchronization between EEG and ET streams, we extracted fixation-related potentials (FRPs) during first-pass reading. Figure 6 shows the resulting FRPs for two representative electrodes (PO8 and Cz), along with scalp topographies at selected time points. Both the FRP waveforms and spatial voltage distributions are consistent with results reported in ZuCo (Hollenstein et al., 2018; 2020), supporting accurate cross-modal alignment in ROAMM. FRPs across all electrodes during both normal reading and MW periods are provided in Appendix Figure 7.

We additionally evaluated the reliability of MW onset annotations. In Sun et al. (2026), we showed that incorporating MW onset information significantly improves the performance of logistic regression models trained to detect MW from eye-tracking features. A sliding-window analysis replicated prior findings of reduced fixation rate during MW episodes, while further demonstrating that these changes emerge near the reported MW onset. This alignment be-

---

[1]Fixations are intervals during which the eyes remain stationary on a word.

[2]Saccades are fast eye movements that reposition gaze between words or regions of text during reading.

tween subjective reports and objective eye-tracking metrics provides evidence for the validity of the annotations. Together, these results suggest that the ReMind paradigm provides reliable temporal localization of MW episodes, supporting the precision of ROAMM's attention-state labels.

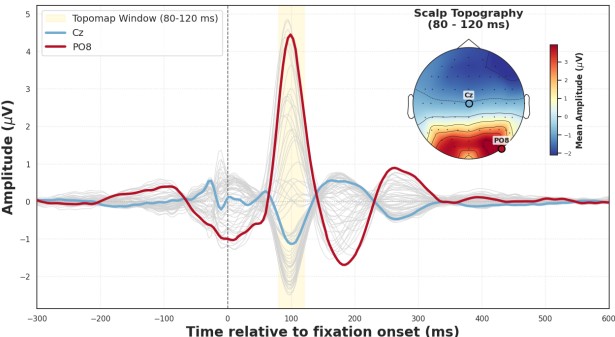

*Figure 6.* Group average fixation-related potentials during first-pass reading, with scalp topographies averaged over 80–120 ms after fixation onset. FRPs for PO8 and Cz are highlighted, and the locations of these electrodes are marked on the topographic map.

## 4. Reference Evaluations

### 4.1. Mind-Wandering Detection

**Feature Engineering.** We extracted ET and EEG features using non-overlapping 2-second windows during first-pass reading. For ET, we computed a standard set of features that have been shown to work well for MW detection in prior studies, including fixation count, mean fixation duration, pupil size during fixations, saccade count, mean saccade duration, mean saccade amplitude, peak saccade velocity, and subject-normalized pupil size (Reichle et al., 2010; Faber et al., 2017).

For EEG, we computed power spectral density (PSD) features in eight frequency bands: theta1 (4.0–6.0 Hz), theta2 (6.5–8.0 Hz), alpha1 (8.5–10.0 Hz), alpha2 (10.5–13.0 Hz), beta1 (13.5–18.0 Hz), beta2 (18.5–30.0 Hz), gamma1

(30.5–40.0 Hz), and gamma2 (40.0–49.5 Hz). These PSD features are commonly used for cognitive state decoding (e.g., sleep staging and emotion-related tasks), and our band definitions were inspired by the ZuCo dataset (Hollenstein et al., 2018). We computed PSD for all 64 channels, resulting in 512 EEG features per window ($64 \times 8$).

Each 2-second window was labeled using the sample-level MW annotations within that window. To keep labels clean, we excluded any windows that contained both MW and non-MW samples (i.e., windows spanning MW onset or offset). Because MW windows were much less frequent than non-MW windows, we balanced the dataset by downsampling the non-MW class to match the MW class size, performed within each subject. We standardized features and applied principal component analysis (PCA) using a scikit-learn pipeline to avoid data leakage.

**Model Selection.** We then trained supervised MW classifiers using seven common models from scikit-learn with default hyperparameters: logistic regression, linear SVC, RBF SVC, random forest, gradient boosting, $k$-nearest neighbors, and a multilayer perceptron. We focused on supervised models because they not only perform well but also make it easier to interpret which EEG and eye-tracking features differ between MW and attentive reading.

**Leave-One-Subject-Out Evaluation.** To evaluate generalization to unseen participants, we adopted a leave-one-subject-out (LOSO) evaluation protocol, training on all subjects except one and evaluating on the held-out subject. Predictions from all folds were aggregated to compute overall metrics including accuracy, precision, F1, and AUROC.

We emphasize that careful evaluation design is critical for MW decoding from time-series data such as EEG. MW annotations are temporally correlated, with contiguous spans of reading labeled as MW, while EEG signals themselves contain strong temporal structure. As a result, arbitrary train/test splits can introduce severe temporal leakage, allowing models to achieve artificially inflated performance by exploiting temporal continuity rather than genuine MW-related neural activity. In addition, arbitrary splits may

*Table 3.* MW classification performance under LOSO evaluation. Significance was assessed using 1,000 subject-level bootstrap iterations while preserving LOSO splits. P-values were computed against a 0.5 chance baseline and FDR-corrected across models and modalities for each metric. Bold values indicate statistically significant results after FDR correction ($p < 0.01$). Underlined values indicate the best-performing modality for each model and metric.

| Model | Accuracy | | | Precision | | | F1 | | | AUROC | | |
|---|---|---|---|---|---|---|---|---|---|---|---|---|
| | ET | EEG | ET+EEG | ET | EEG | ET+EEG | ET | EEG | ET+EEG | ET | EEG | ET+EEG |
| **LogReg** | **0.567** | **0.554** | **0.573** | **0.576** | **0.557** | **0.579** | 0.538 | 0.538 | 0.556 | **0.592** | **0.569** | **0.599** |
| **Linear SVC** | **0.567** | **0.552** | **0.572** | **0.576** | **0.556** | **0.578** | 0.539 | 0.536 | 0.554 | **0.592** | **0.566** | **0.599** |
| **Gradient Boosting** | **0.571** | **0.553** | **0.566** | **0.568** | **0.558** | **0.571** | 0.578 | 0.530 | 0.548 | **0.588** | **0.572** | **0.587** |
| **RBF SVC** | 0.569 | 0.551 | 0.572 | 0.570 | 0.553 | 0.576 | 0.567 | 0.538 | 0.559 | 0.586 | 0.569 | 0.600 |
| **kNN** | **0.535** | 0.528 | **0.537** | **0.534** | 0.525 | **0.534** | 0.539 | 0.546 | 0.556 | **0.549** | 0.544 | **0.557** |
| **MLP** | **0.560** | **0.549** | **0.547** | **0.560** | **0.551** | **0.546** | 0.559 | 0.539 | 0.549 | **0.582** | **0.558** | **0.560** |
| **Random Forest** | 0.549 | 0.566 | 0.576 | 0.549 | 0.585 | 0.592 | 0.550 | 0.509 | 0.533 | **0.565** | **0.590** | **0.609** |

enable models to leverage subject-specific neural signatures, further confounding evaluation. To illustrate this issue, we additionally trained subject-specific classifiers using 10-fold train/test splits on each participant's own data and observed substantially higher performance compared to LOSO evaluation (Appendix Figure 8). By adopting LOSO evaluation, we mitigate both temporal and subject-identity leakage, providing a more conservative and realistic estimate of cross-subject generalization performance.

**Detection Results.** We trained three versions of the MW classifier using (1) ET features only, (2) EEG features only, and (3) a hybrid feature set combining both modalities. Results are summarized in Table 3, with per-subject model performance reported in the Appendix. For most classifiers, combining EEG and eye-tracking features improved MW detection performance compared to using either modality alone. The best-performing cross-subject model was a random forest classifier, achieving an AUROC of 0.609 (chance = 0.5). For comparison, the best-performing within-subject model achieved an AUROC of 0.92 for one participant, with a mean AUROC of 0.72 across participants (Appendix Figure 8). To better understand which features contributed most to cross-subject performance, we examined random forest feature importance; results are reported in Appendix Table 8.

### 4.2. EEG-to-Text Decoding

Prior work has demonstrated promising EEG-to-text decoding performance on the ZuCo datasets. Here, we test whether EEG-to-text decoding remains feasible in ROAMM, which involves more naturalistic and less controlled free-viewing, page-level reading. To enable a direct comparison, we extract EEG features following the ZuCo preprocessing pipeline (Hollenstein et al., 2018): we compute word-level power spectral density (PSD) features across eight frequency bands for all 64 channels during each fixation, and group these features into sentence-level sequences. We then adopt the EEG-to-text decoding approach from Wang & Ji (2022), which uses an EEG encoder to produce embeddings that are used to fine-tune a pretrained BART model, and whose results on ZuCo have served as a widely used reference baseline in subsequent work (Zhou et al., 2024; Liu et al., 2024; Wang et al., 2024). We follow the original model design and training procedure, modifying only the input dimensionality to match our feature representation.

One difference in dataset preparation concerns how multiple fixations on the same word are handled. In (Wang & Ji, 2022), EEG features from multiple fixations on the same word are treated as separate inputs. In ROAMM, all 44 participants read the same materials, and allowing multiple fixations per word would substantially inflate the number of fixation events per word in each sentence sequence. To

avoid this imbalance and reduce redundancy, we average EEG features across fixations on the same word. Because MW episodes are typically short, the number of MW-related fixations (27,894) is much smaller than that of non-MW fixations (365,450). To prevent this sample-size disparity from confounding the effect of averaging EEG features, we downsampled non-MW fixations to match the number of MW fixations when constructing the MW and NON-MW subsets. For dataset ALL, we averaged samples across both MW and NON-MW periods without additional balancing. As a result, the different datasets used for EEG-to-text decoding contain comparable numbers of training samples, as reported in Table 4.

We train and evaluate decoding models on three subsets of ROAMM: (1) all segments, (2) non-MW segments only, and (3) MW segments only. For each subset, we use an 80/10/10 split for training, development, and evaluation based on unique sentences. Dataset statistics and decoding performance (BLEU and ROUGE-1 scores) across attention conditions are reported in Table 4.

*Table 4.* EEG-to-text decoding performance across ROAMM attention subsets, compared with ZuCo datasets, whose results are taken from prior work using the same decoding architecture (Wang & Ji, 2022).

| DATASET | # TRAIN | BLEU-N | | ROUGE-1 F1 |
|---|---|---|---|---|
| | | $N{=}1$ | $N{=}2$ | |
| MW | 6063 | 0.057 | 0.009 | 0.094 |
| ALL | 6063 | 0.168 | 0.054 | 0.241 |
| NON-MW | 6063 | 0.189 | 0.071 | 0.258 |
| ZUCO | 10710 | 0.401 | 0.231 | 0.301 |

## 5. Discussion

ROAMM was designed to capture real-world reading behavior and corresponding physiological signals in naturalistic settings by combining co-registered EEG and ET with time-resolved MW annotations. We first evaluated MW detection using supervised classifiers and found that combining EEG and ET features improved performance for most models, supporting the view that MW has measurable and complementary signatures in both ocular behavior and neural dynamics (Steindorf & Rummel, 2019; Kam et al., 2022). Under LOSO evaluation, the best-performing cross-subject model achieved an AUROC of 0.609. Notably, we previously reported an ET-only MW classification AUROC of 0.659 in Sun et al. (2026). However, that analysis used windows spanning the full MW episode duration, whereas the present work uses fixed 2 s windows to better reflect realistic deployment settings such as continuous attention monitoring or MW notification systems. Because ET features are computed over temporal intervals, longer windows generally produce more stable feature estimates and can therefore lead to improved classification performance.

Importantly, despite the increased variability and noise inherent in naturalistic reading, MW classification performance measured by AUROC on ROAMM (cross-subject = 0.609; mean within-subject = 0.72) remains comparable to results reported in prior studies using more controlled paradigms. For comparison, Jin et al. (2019) reported a cross-subject score of approximately 0.60; Dong et al. (2021) reported cross-subject and within-subject scores of 0.613 and 0.715, respectively; and Tang et al. (2025) reported corresponding scores of 0.56 and 0.68. Together, these comparisons suggest that within-subject evaluation generally produces substantially higher performance than LOSO evaluation in MW classification tasks. These differences highlight the importance of LOSO evaluation for assessing true cross-subject generalization while mitigating potential temporal and subject-specific data leakage. Overall, these comparisons support both the robustness and ecological validity of ROAMM as a benchmark for MW decoding under realistic reading conditions.

To better understand the drivers of model performance, we examined feature importance for the random forest classifier. The most informative features reflected both ocular and neural signatures of MW. ET measures such as fixation and saccade counts contributed strongly to the most discriminative components. This is consistent with the perceptual decoupling theory (Smallwood & Schooler, 2006) that during MW, attentional resources are allocated to internally generated thoughts rather than external stimuli. In line with our previous work, fixation and saccade rates were reduced during MW compared to attentive reading. In addition, incorporating EEG features improved classification performance beyond eye tracking alone. In particular, band-power features in the theta ranges contributed prominently, supporting the idea that frequency-specific EEG activity reflects distinct cognitive and attentional states (Jin et al., 2019; Rodriguez-Larios & Alaerts, 2020).

Second, we evaluated EEG-to-text decoding performance on ROAMM using the same architecture and training procedure as prior work on ZuCo (Wang & Ji, 2022) in order to assess how well existing decoding approaches generalize to more naturalistic reading conditions. Overall decoding performance was lower than the results reported on the ZuCo datasets, which is expected given that ROAMM involves a more complex and naturalistic reading setting with full pages of continuous text (including multiple sentences and paragraphs), whereas ZuCo (Hollenstein et al., 2018; 2020) uses a more controlled sentence-level paradigm. In multi-page comprehension reading, participants may also adopt more variable strategies (e.g., skipping ahead, revisiting key phrases, or scanning for informative content) (Rayner, 1998), which can weaken the alignment between fixation order and incremental sentence-level linguistic processing. In contrast, sentence-level reading provides a more constrained

context in which neural responses are more tightly coupled to processing a single sentence, potentially yielding cleaner supervision for decoding models.

Despite the lower absolute decoding performance, ROAMM reveals a clear and practically important effect of attention. Models trained and evaluated on MW-labeled segments show a substantial reduction in decoding performance, which supports our hypothesis that attention state critically shapes neural language representations. We argue that this effect reflects the hierarchical functional organization of the reading (Dehaene & Cohen, 2011; Chen et al., 2019). Prior work shows that individuals who are more prone to being off-task exhibit greater decoupling between default mode network (DMN) regions involved in reading and primary visual cortex (Zhang et al., 2022). Under this account, MW may preserve lower-level sensory processing of visual input while disrupting higher-level integration between perceptual and semantic/language systems, thereby weakening the neural representations most relevant for linguistic decoding. More broadly, these results highlight MW as a key confound for neural language decoding and motivate attention-aware decoding approaches that explicitly model cognitive state rather than treating all reading segments as equally informative.

To conclude, ROAMM presents novel utility to the machine learning and neuroscience communities in three major ways. First, ROAMM is the first EEG dataset that provides precise estimates of both the onset and offset times of MW. This enables time-resolved studies of MW and its effects on semantic processing and eliminates assumptions about MW duration. Second, because the dataset includes ET data observed during reading, classifiers can use multimodal data streams in these studies; we demonstrate that these are complementary when predicting MW. Finally, ROAMM provides an opportunity for EEG-to-text decoding that includes naturalistic eye movements and MW annotations, providing a more realistic and holistic dataset for benchmarking how semantic information can be decoded from neural activity in the real world.

## 6. Limitations and Future Work

There are several limitations to ROAMM that suggest important directions for future work. First, although we assessed page-level reading comprehension using multiple-choice questions, this measure provides only a coarse proxy for true comprehension. Many questions required recall of specific factual details (e.g., numbers or names), which may be missed even when a participant has a strong overall understanding of the narrative. Incorporating more inferential or integrative comprehension measures could yield a more complete characterization of reading success and its relationship to attention fluctuations (Ghanem et al., 2022).

Second, our MW labels do not distinguish the content or subtype of mind-wandering. Participants were instructed to report any internal thought that disrupted their reading flow, which likely includes a broad range of off-task experiences (Girardeau et al., 2022). For example, MW could involve thoughts entirely unrelated to the text, or it could be cued by the text itself, for example, autobiographical memories triggered by the story. These categories may reflect different cognitive mechanisms and neural signatures (Gross et al., 2020). Collecting additional annotations about MW content could enable finer-grained analyses and support classifiers that differentiate spontaneous versus stimulus-dependent MW.

Third, MW classification and EEG-to-text decoding performance were not perfect. One common factor that may constrain performance in both tasks is substantial inter-individual variability in ocular and neural dynamics (Traxler et al., 2012; Kucyi et al., 2024), as well as differences in functional network organization (Jangraw et al., 2023). Specifically, for MW classification, EEG provided only limited gains over ET alone. We hypothesize that this reflects physiological coupling between modalities, as Henke et al. (2023) reported correlations between neural oscillations and saccadic behavior during reading. With standard feature engineering approaches and baseline classifiers, shared variance between EEG and ET signals may therefore be captured redundantly. For EEG-to-text decoding, we strictly followed the original baseline implementation by averaging EEG features across multiple fixations (Wang & Ji, 2022) to maintain comparability with existing benchmarks. However, this averaging procedure likely discards temporally fine-grained neural information that may be important for decoding linguistic representations during continuous reading.

Consequently, ROAMM provides substantial room for future methodological advances. For MW detection, it will be valuable to explore neural architectures specifically designed for time-series and EEG data (Lawhern et al., 2018; Goldstein et al., 2019; Cui et al., 2021). Beyond potentially improving robustness to individual variability, such models may uncover non-linear neural signatures not captured by ET features alone, thereby better leveraging complementary information across modalities. For EEG-to-text decoding, recent methods have reported improved performance on ZuCo compared to the baseline model adopted here (Zhou et al., 2024; Liu et al., 2024; Wang et al., 2024). Applying these newer methods to ROAMM may yield improved decoding performance and greater robustness under naturalistic reading conditions.

Beyond these methodological opportunities, ROAMM also opens new research opportunities and directions. A key contribution of ROAMM is its span-level MW annotations,

which enable direct investigation of the alignment and divergence between human and machine language processing. For example, ROAMM makes it possible to compare human attention dynamics with attention patterns produced by large language models (LLMs) by correlating MW labels with model-derived token importance or attention scores, rather than relying solely on indirect proxies of human attention such as gaze behavior (Mouratidi & Poesio, 2025; Lopez-Cardona et al., 2025).

Finally, together with ZuCo (Hollenstein et al., 2018; 2020), ROAMM contributes to a small but growing set of simultaneous EEG and ET datasets for reading. Expanding this ecosystem with additional naturalistic datasets may enable larger-scale multimodal modeling of human reading, including foundation models that integrate EEG, eye movements, and language content to predict comprehension and other downstream cognitive outcomes.

## 7. Conclusion

We introduced ROAMM, a multimodal dataset of naturalistic multi-page reading that combines co-registered EEG and ET with span-level MW annotations. ROAMM enables systematic study of how attention fluctuations shape behavioral and neural dynamics during realistic reading. Using established feature pipelines and supervised classifiers, we showed that MW can be detected at fine temporal resolution and that multimodal EEG and ET features improve detection performance under LOSO evaluation. We further demonstrated that MW is a significant confound for neural language decoding by showing that the performance of EEG-to-text decoding decreases when MW-labeled segments are included. Overall, ROAMM offers a unique resource and evaluation protocol for multimodal MW detection and neural decoding, and motivates future attention-aware decoding models for robust cognitive and BCI applications in real-world settings.

## Impact Statement

This paper presents work whose goal is to advance the field of Machine Learning. There are many potential societal consequences of our work, none of which we feel must be specifically highlighted here.

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

# A. ReMind Task Paradigm

## A.1. Mind-Wandering Definition

Before the first run, participants received detailed task instructions, including an explicit definition of mind-wandering (MW). MW was defined as any thought that interfered with remembering the recently read text. Example thoughts included completely unrelated thoughts (e.g., "What did I have for lunch?"), loosely text-related thoughts (e.g., "I wonder where this town is?"), brief self-reflective thoughts (e.g., "I'm tired."), blank states (e.g., "thinking about nothing"), or even task-relevant but distracting thoughts (e.g., "I should remember that.").

## A.2. Comprehension Questions

To encourage attentive reading and assess comprehension, we designed one multiple-choice question for each page. Questions were constructed such that they could only be answered correctly if participants attended to the content of that specific page. For example, after reading the sentence "...Special effects were introduced and film continuity, involving action moving from one sequence into another, began to be used...", participants were asked: "*What does film continuity mean?*" with answer choices including: (A) the plot of the film was fluent; (B) the film was played continuously without interruption; (C) action moving from one sequence into another; (D) motion pictures were produced without sound; and (E) I am not sure.

After completing each article, participants answered all page-level comprehension questions in randomized order. They then rated the overall understandability of the article and reported their familiarity with the topic using 5-point Likert scales.

## A.3. Reliable MW Onset

In Sun et al. (2026), we additionally analyzed how eye-tracking measures changed over time relative to reported MW onset using a sliding-window framework. Fixation rate showed a reliable decrease beginning shortly before MW onset and remained lower throughout the MW interval. Fixation dispersion also declined following MW onset before stabilizing several seconds later. Pupil size exhibited a gradual increase after MW onset, consistent with changes in attentional engagement and arousal during MW. Together, these results demonstrate that multiple eye-tracking features exhibit systematic temporal changes surrounding reported MW onset, supporting the reliability of the ReMind onset annotations.

# B. ROAMM Dataset

## B.1. Participant Demographics

*Table 5.* Participant demographics for the ROAMM dataset (N=44).

| Subject ID | Age | Gender | Handedness | ADHD |
|---|---|---|---|---|
| 10014 | 21 | female | right | No |
| 10052 | 29 | female | right | Yes |
| 10059 | 64 | female | right | No |
| 10073 | 19 | non-binary | left | No |
| 10081 | 27 | female | right | No |
| 10084 | 34 | male | right | Yes |
| 10085 | 28 | female | left | No |
| 10089 | 26 | female | left | No |
| 10094 | 22 | female | right | No |
| 10100 | 30 | female | right | No |
| 10103 | 19 | male | right | No |
| 10110 | 18 | female | right | No |
| 10111 | 22 | non-binary | right | Yes |
| 10115 | 18 | female | right | No |
| 10117 | 20 | female | right | No |
| 10121 | 28 | female | right | No |
| 10125 | 22 | male | right | No |
| 10138 | 20 | non-binary | right | No |
| 10139 | 19 | female | right | No |
| 10141 | 19 | female | right | No |
| 10144 | 19 | female | right | No |
| 10145 | 19 | non-binary | right | No |
| 10148 | 36 | male | left | No |
| 10153 | 18 | female | right | No |
| 10156 | 18 | female | right | No |
| 10158 | 21 | female | right | Yes |
| 10159 | 27 | female | right | Yes |
| 10160 | 20 | female | right | Yes |
| 10165 | 18 | female | right | No |
| 10173 | 18 | female | right | No |
| 10177 | 18 | female | right | No |
| 10178 | 22 | male | right | No |
| 10180 | 19 | male | right | Yes |
| 10181 | 18 | female | left | No |
| 10183 | 20 | non-binary | right | No |
| 10185 | 18 | female | right | No |
| 10186 | 19 | male | right | No |
| 10188 | 19 | male | right | No |
| 10192 | 18 | female | right | No |
| 10195 | 21 | female | right | No |
| 10196 | 20 | female | left | No |
| 10197 | 20 | male | right | No |
| 10200 | 22 | female | left | No |
| 10202 | 20 | female | right | Yes |

## B.2. Dataset Scale

The ROAMM dataset is a rich multimodal resource comprising over 46 million samples collected from 44 participants, totaling more than 50 hours of synchronized EEG and eye-tracking recordings. Within this corpus, approximately 26 million samples ($\approx$ 30 hours) correspond to first-pass reading, with word-level fixation metadata provided for each sample, including approximately 2.2 aggregate hours of annotated mind-wandering data.

*Table 6.* ROAMM dataset scale.

| DATA TYPE | #SAMPLES | TIME | SUBJECT AVG TIME |
|---|---|---|---|
| TOTAL RECORDING | 46,371,584 | 50H 18M | 1H 8M |
| FIRST-PASS READING | 26,691,014 | 28H 57M | 39M |
| MIND-WANDERING | 2,042,601 | 2H 12M | 3M |
| FIXATIONS | 38,316,711 | 41H 34M | 56M |
| SACCADES | 9,323,085 | 10H 6M | 13M |
| BLINKS | 2,287,380 | 2H 28M | 3M |

## B.3. Dataset Structure

ROAMM is organized as synchronized time-series data to facilitate downstream machine learning analyses. EEG recordings and moment-to-moment eye-tracking signals, such as gaze position and pupil size, are temporally aligned onto a shared sampling timeline. Eye events that span temporal intervals, including fixations, saccades, and blinks, are expanded into the time-series representation. For example, a fixation lasting 200 ms corresponds to approximately $(200/1000) \times 256 \approx 51$ consecutive rows in the synchronized data after downsampling to 256 Hz, with each row containing the associated fixation metadata. The same representation is applied to saccades and blinks, enabling direct alignment between continuous EEG activity and eye-event annotations.

*Table 7.* Column metadata for synchronized ROAMM EEG and eye-tracking files.

| Column(s) | # Cols | Unit | Description |
|---|---|---|---|
| Fp1–O2 | 64 | V | 64 EEG channel voltage values using BioSemi 10–20 electrode labels. |
| time | 1 | s | Timestamp aligned to the synchronized EEG time base. |
| sfreq | 1 | Hz | Sampling frequency after synchronization/downsampling. |
| first_pass_reading | 1 | binary | Whether the sample belongs to first-pass reading. |
| page_num | 1 | index | Page number within the current article. |
| page_start, page_end | 2 | s | Start and end timestamps of the current page. |
| page_dur | 1 | s | Duration of the current page. |
| is_mw | 1 | binary | Whether the sample is labeled as mind-wandering. |
| mw_onset, mw_offset | 2 | s | Estimated onset and offset timestamps of the MW episode. |
| mw_dur | 1 | s | Duration of the MW episode. |
| run_num | 1 | index | Article/run number within the session. |
| story_name | 1 | text | Name of the Wikipedia article being read. |
| is_fix | 1 | binary | Whether the sample occurs during a fixation. |
| fix_[L/R]_eye | 2 | text | Eye associated with the fixation event. |
| fix_[L/R]_tStart, fix_[L/R]_tEnd | 4 | s | Start and end timestamps of the fixation. |
| fix_[L/R]_duration | 2 | ms | Duration of the fixation. |
| fix_[L/R]_xAvg, fix_[L/R]_yAvg | 4 | px | Average fixation position on the screen. |
| fix_[L/R]_pupilAvg | 2 | arbitrary unit | Average pupil area during the fixation. |
| fix_[L/R]_fixed_word | 2 | text | Word fixated during the event. |
| fix_[L/R]_fixed_word_key | 2 | UUID | Unique key linking the fixation to a word in the story file. |
| is_blink | 1 | binary | Whether the sample occurs during a blink. |
| blink_[L/R]_eye | 2 | text | Eye associated with the blink event. |
| blink_[L/R]_tStart, blink_[L/R]_tEnd | 4 | s | Start and end timestamps of the blink. |
| blink_[L/R]_duration | 2 | ms | Duration of the blink. |
| is_sacc | 1 | binary | Whether the sample occurs during a saccade. |
| sacc_[L/R]_eye | 2 | text | Eye associated with the saccade event. |
| sacc_[L/R]_tStart, sacc_[L/R]_tEnd | 4 | s | Start and end timestamps of the saccade. |
| sacc_[L/R]_duration | 2 | ms | Duration of the saccade. |
| sacc_[L/R]_xStart, sacc_[L/R]_yStart | 4 | px | Starting screen position of the saccade. |
| sacc_[L/R]_xEnd, sacc_[L/R]_yEnd | 4 | px | Ending screen position of the saccade. |
| sacc_[L/R]_ampDeg | 2 | degrees | Saccade amplitude in visual angle. |
| sacc_[L/R]_vPeak | 2 | deg/s | Peak saccade velocity. |
| tSample | 1 | s | Original eye-tracking sample timestamp. |
| LX, LY, RX, RY | 4 | px | Raw left/right gaze coordinates. |
| LPupil, RPupil | 2 | arbitrary unit | Raw left/right pupil measurements. |
| blink_interp_[LX/LY/RX/RY] | 4 | px | Blink-interpolated gaze coordinates. |
| blink_interp_[LPupil/RPupil] | 2 | arbitrary unit | Blink-interpolated pupil measurements. |

## B.4. Group Fixation-Related Potentials

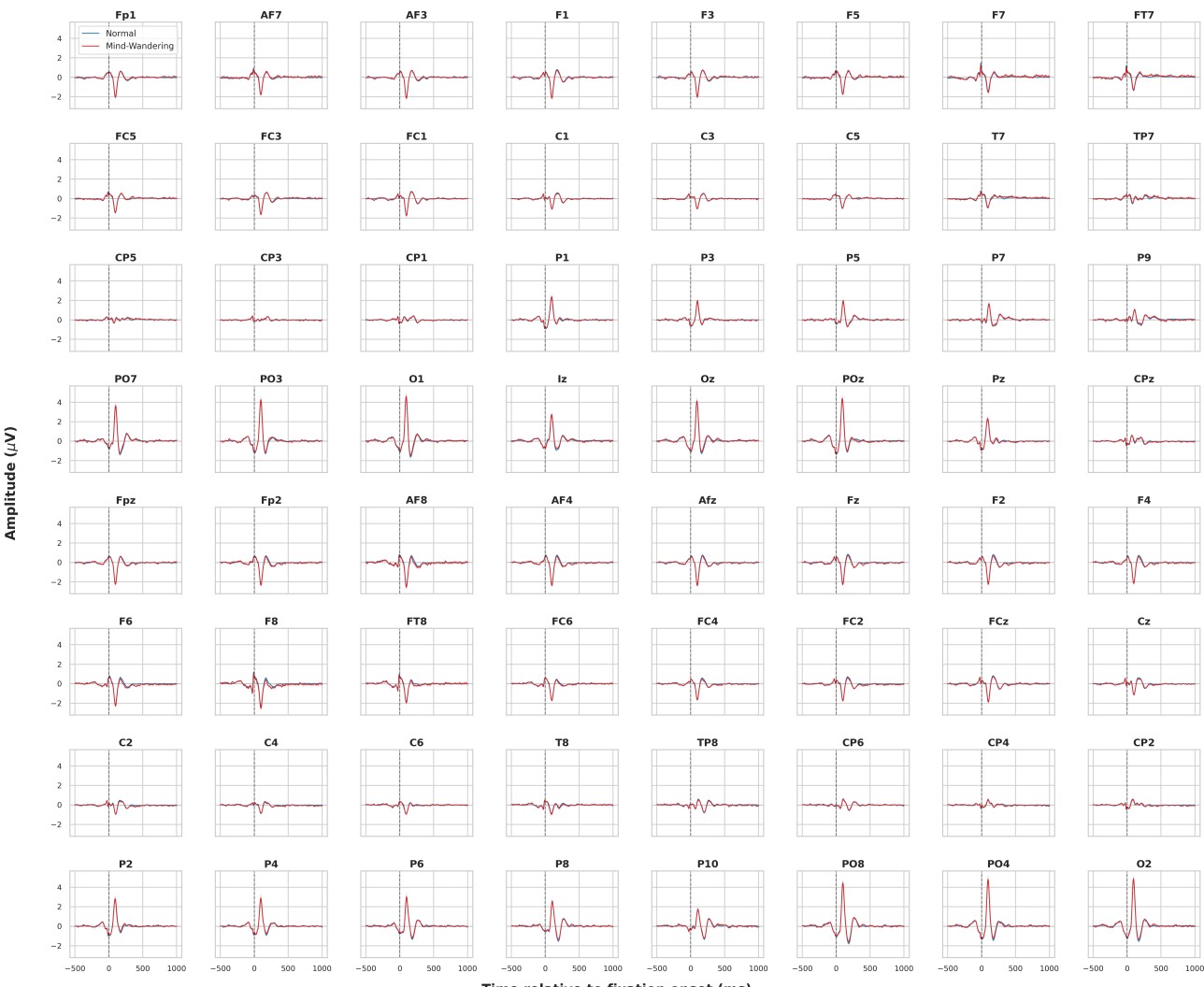

*Figure 7.* Group-average fixation-related potentials (FRPs) during normal reading and MW across all 64 electrodes. FRPs were directly epoched from the synchronized EEG and ET recordings. Shaded regions indicate standard error across participants ($N = 44$).

## C. Mind-Wandering Classifier

### C.1. Within-Subject Classifier Performance

To illustrate the conservative nature of LOSO evaluation, we additionally trained subject-specific MW classifiers using each participant's own data. To maintain class balance, the majority class was downsampled prior to training. We used 10-fold cross-validation for train/test splitting, while keeping the remaining pipeline consistent with the group-level classifier analysis, including the use of multiple modalities (ET, EEG, and ET+EEG) and classifier types. For each participant, we report the best-performing classifier across all evaluated models and modality combinations.

The best-performing within-subject classifier was a Linear SVC trained on the combined EEG and ET feature set, achieving an AUROC of 0.92. Across participants, AUROC values ranged from 0.57 to 0.92, with a mean AUROC of 0.72. Compared to LOSO evaluation, these substantially higher within-subject results further demonstrate the importance of subject-independent evaluation and highlight the potential impact of temporal and subject-specific leakage in MW decoding tasks.

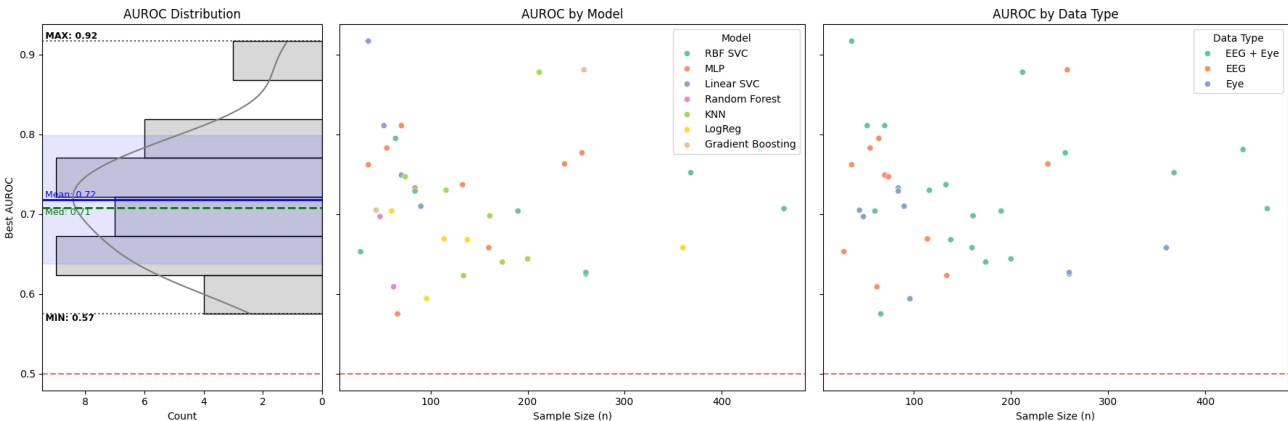

*Figure 8.* Subject-specific MW classification performance across participants. The left panel shows the distribution of the best within-subject AUROC scores. Scatter plots show the relationship between AUROC and sample size, with points colored by classifier and modality type.

### C.2. Random Forest Feature Importance

To examine feature importance, we trained a random forest classifier following the same pipeline using data pooled across all participants. Feature importance scores were extracted directly from the trained classifier. Because dimensionality reduction was performed using PCA, we analyzed the original EEG and eye-tracking features by identifying those with the highest loadings on the five most important principal components. Table 8 summarizes the top contributing original features for each of these components.

*Table 8.* Top contributing original features to the five most important PCA components.

| PC | Importance | Top Features |
|---|---|---|
| PC33 | 0.0148 | fix_count, sacc_count, fix_R_duration, F6_theta1 |
| PC34 | 0.0127 | fix_count, sacc_count, fix_R_duration, CP5_theta2 |
| PC61 | 0.0125 | T8_theta1, FC6_theta1, PO8_beta2, CP3_theta1 |
| PC31 | 0.0125 | fix_count, sacc_count, fix_R_duration, F6_theta1 |
| PC23 | 0.0115 | Pz_theta1, P2_theta1, PO4_theta1, T7_alpha1 |

