# OpenReview forum: "ROAMM: A Benchmark Dataset for Multimodal Human Attention Decoding and EEG-to-Text Modeling During Naturalistic Reading"
_ICML.cc/2026/Conference — ICML 2026 regular_

### Official Review · Reviewer_7pkg · 2026-03-02

**Soundness:** 2
**Presentation:** 3
**Significance:** 2
**Originality:** 2
**Overall Recommendation:** 4
**Confidence:** 4

**Summary:**

The authors introduce ROAMM, a large-scale multimodal dataset comprising 50 hours of co-registered EEG and eye-tracking (ET) data from 44 participants during naturalistic, multi-page reading. The authors propose a standardized evaluation protocol for mind-wandering (MW) detection, achieving an AUROC of 0.609 under leave-one-subject-out (LOSO) evaluation. Additionally, the authors demonstrate that the inclusion of MW-labeled segments degrades performance in EEG-to-text decoding tasks, positioning ROAMM as a benchmark for developing attention-aware BCI models.

**Compliance With Llm Reviewing Policy:**

Affirmed.

**Final Justification:**

The rebuttal addressed my concerns about the AUROC results and EEG feature aggregation; however, the utility of the modalities remains a concern.

**Key Questions For Authors:**

1- Since ET alone performs competitively with the hybrid ET+EEG model, what empirical evidence (e.g., statistical significance tests) confirms that EEG adds unique, non-redundant information for MW detection?

2- You downsampled the majority (non-MW) class. Have you evaluated synthetic data generation or class-weighting to preserve temporal information, and how did these impact the AUROC?

3- Your decoding performance on "ALL" and "NON-MW" subsets is lower than the reported results on the ZuCo datasets. Beyond the "naturalistic vs. controlled" distinction, do you attribute this primarily to the averaging of EEG features per word, and have you tested if this lossy preprocessing disproportionately affects EEG-based decoding versus ET-based signals?

4- Given the modest AUROC (0.609) under LOSO, what are the primary barriers to better inter-subject generalization? Is this driven more by individual differences in neural architecture or by the inherent noise of self-reported MW?

**Limitations:**

Yes.

**Strengths And Weaknesses:**

# Soundness

The dataset collection pipeline is technically sound and robust, utilizing established preprocessing tools like EEGLAB and the unfold toolbox for fixation-related potential (FRP) extraction. However, the empirical evidence for the utility of a multimodal approach is weak. As shown in Table 3, the performance gains from adding EEG to ET features are marginal, and in several instances, ET-only models outperform the hybrid ET+EEG configuration. Without formal significance testing, the authors fail to demonstrate that the increased complexity of this multimodal dataset provides a tangible benefit over using ET alone.

# Presentation

The paper is generally well-structured and clear. The narrative successfully positions ROAMM as a more naturalistic extension of the ZuCo benchmarks. To improve clarity, the authors should better justify the "EEG-to-text" preprocessing choices, specifically, the decision to average EEG features across fixations on the same word, which risks discarding high-temporal-resolution neural information compared to alternative sequence-modeling approaches.

# Significance
This work addresses a critical gap by providing time-resolved MW annotations in a naturalistic setting. The finding that MW acts as a confound for neural decoding is significant and likely to influence future BCI research. However, the reported AUROC of 0.609 is barely above chance (0.50). While the authors argue this is comparable to existing literature, it suggests that the current feature set (PSD and basic ET metrics) may be insufficient to capture the nuances of MW in naturalistic settings.

# Originality
The primary contribution is the dataset's unique design: combining naturalistic, multi-page reading with precise, retrospectively-reported, word-level MW onset/offset labels.

---

> ### Author Rebuttal · Authors · 2026-03-28
>
> We thank the reviewer for their thoughtful critique and for recognizing ROAMM as a more naturalistic extension of benchmarks like ZuCo. We address the key questions below.
>
> ## Modality Utility ##
> Our preliminary results show that EEG provides limited gains over ET alone for MW classification. We hypothesize this reflects physiological coupling between modalities: Henke et al. (2023) shows correlations between neural oscillations and saccades during reading. With standard feature engineering and baseline models, shared variance is likely captured redundantly. We will clarify this limitation.
>
> This modest baseline offers an opportunity for new approaches:  ROAMM enables deep learning approaches that operate on raw EEG, which may uncover non-linear neural signatures not captured by ET features. Moreover, the value of ROAMM extends beyond MW classification: co-registered EEG is essential for tasks such as EEG-to-text decoding, which require precise neural–linguistic alignment via fixation information.
>
> ## Class Imbalance ##
> We are cautious about synthetic augmentation for high-dimensional biological signals. Simple oversampling risks overfitting to specific MW instances. While downsampling reduces data, it provides a conservative and biologically grounded baseline.
>
> We appreciate the suggestion of class weighting and will include it in the revision
>
> ## Word-Level EEG Feature Aggregation ##
> We share the reviewer’s concern regarding potential loss of temporal information when averaging EEG features across multiple fixations. We adopted the word-level aggregation approach (averaging fixations) used by Wang & Ji (2024), as documented in the ZuCo processing scripts in their OSF repository (https://osf.io/q3zws/files/gan9v), to maintain comparability with existing benchmarks. While this introduces a trade-off, it ensures consistency with prior work. We will explicitly acknowledge this limitation in the manuscript.
>
> ## Performance Difference Between ROAMM and ZuCo ##
> We attribute the performance differences primarily to ecological validity. Naturalistic settings introduce lower signal-to-noise ratios compared to controlled laboratory experiments, which can negatively impact decoding performance.
>
> In terms of hardware, ZuCo uses a 128-channel EEG system, which provides higher spatial resolution. While we believe that a 64-channel system offers sufficient spatial sampling for most EEG analyses, it is possible that for decoding tasks, particularly those requiring fine-grained neural pattern discrimination, the additional spatial resolution may offer performance benefits.
>
> ## Inter-Subject Generalization ##
> The modest LOSO AUROC (0.609) reflects two key challenges: noisy MW labels and substantial inter-subject variability. Mind-wandering is a spontaneous, internal, and often unconscious cognitive state, and current detection paradigms inherently introduce "ground-truth" noise. Furthermore, our individual-level classification results (mean AUROC = 0.72, range = [0.57, 0.92]) show substantial variability across participants, suggesting that MW signatures are not uniform across the population: https://anonymous.4open.science/r/roamm_ml-CB37/individual_MW_classifier_perf.png
>
> Although some studies report higher performance, our AUROC and the gap between within- and cross-subject performance are consistent with prior work. For example:
> - Jin et al. (2019): Average LOSO AUC ≈ 0.60
> - Dong et al. (2021): LOSO AUC = 0.613 vs. within-subject AUC = 0.715
> - Tang et al. (2025): LOSO AUC = 0.56 vs. within-subject AUC = 0.68
>
> Despite these challenges, we maintain that LOSO provides the most rigorous evaluation for this task. As noted by Reviewer fRKr, arbitrary train/test splits can yield confounded results by allowing models to exploit subject-specific neural traits or temporal leakage. Although our preliminary within-subject models achieve higher performance, we intentionally adopt the more conservative LOSO setting to assess true cross-subject generalization rather than subject-identity decoding. We will further clarify how biological variability and label noise limit zero-shot generalization, and note that personalized, longitudinal data collection would likely yield substantial performance improvements for individual-level applications.
>
> Henke, L., Lewis, A. G., and Meyer, L. Fast and slow
> rhythms of naturalistic reading revealed by combined
> eye-tracking and electroencephalography. The Journal
> of Neuroscience, 43(24):4461–4469, May 2023.
>
> Jin, C. Y., Borst, J. P., and van Vugt, M. K. Predict-
> ing task-general mind-wandering with eeg. Cognitive,
> Affective, amp; Behavioral Neuroscience, 19(4):1059–
> 1073, March 2019.
>
> Dong, H. W., Mills, C., Knight, R. T., and Kam, J. W. Y. De-
> tection of mind wandering using eeg: Within and across
> individuals. PLOS ONE, 16(5):e0251490, May 2021.
>
> Tang, S., Jiang, C., and Li, Z. Detecting mind wandering
> via eeg and facial video features. Behavior Research
> Methods, 57(11), October 2025

---

> > ### Author Rebuttal · Reviewer_7pkg · 2026-04-01
> >
> > Thanks for the detailed rebuttal. The addition of the FDR-corrected p-values and the comparison to LOSO results in existing literature effectively addresses my concerns regarding the statistical significance and the 0.609 AUROC baseline. I also appreciate the clarification on using word-level EEG aggregation to maintain consistency with the ZuCo benchmarks.
> >
> > The dataset is a meaningful contribution to the community, and the technical clarifications provided are solid. Given this, I am happy to raise my score. However, since the multimodal utility still appears marginal with current methods, I don't feel comfortable raising the score any further.

---

> > > ### Author Response · Authors · 2026-04-03
> > >
> > > Thank you very much for your thoughtful response and for raising your score. We truly appreciate your time and consideration.
> > >
> > > We understand that the performance gains from multimodal features using traditional supervised machine learning models are currently modest. While this is somewhat expected given the simplicity of these baseline approaches, we share your view that there is substantial room for improvement. One of our primary goals in releasing ROAMM is to enable the community to explore more advanced modeling approaches, such as deep learning, multimodal fusion architectures, and attention-based models, that may better leverage the richness of the dataset and move toward more practical, real-world applications. We hope that ROAMM will serve as a useful benchmark and foundation for developing more sophisticated models capable of capturing complex cognitive dynamics during naturalistic reading.
> > >
> > > Thank you again for your constructive feedback and support!

---

### Official Review · Reviewer_Pqan · 2026-03-13

**Soundness:** 3
**Presentation:** 3
**Significance:** 2
**Originality:** 2
**Overall Recommendation:** 4
**Confidence:** 2

**Summary:**

This paper presents ROAMM, a benchmark dataset containing 50 hours of EEG and eye-tracking recordings from multiple subjects while reading text, with mind-wandering (MW) labels collected via self-report. The readings consist of multi-page articles (unlike prior sentence-level datasets) with an assessment of page comprehension.

The paper conducts two evaluations on this new dataset. The first involves training several classical ML models to classify whether MW occurred using both EEG and ET modalities, as well as each individually, to assess the contribution of each modality. The second evaluation aims to decode the text from the EEG and ET recordings.

The primary contribution is the release of this dataset. Other contributions include the two evaluations mentioned above.

**Compliance With Llm Reviewing Policy:**

Affirmed.

**Final Justification:**

I think this paper is sound; the rebuttal addressed most of my concerns, so I recommend that the paper be accepted.

**Key Questions For Authors:**

•	Do repeated experiments on Table 3 show a significant difference between single and multiple modality training? And isn’t 0.61 AUROC close to random chance?

**Limitations:**

yes

**Strengths And Weaknesses:**

Strengths

Soundness:
The acquisition setup and modality synchronization are all sound. The reading material spans multiple pages and covers a variety of topics, which is a great addition. Apart from concerns related to MW labeling, the paper is generally sound.
Presentation
The paper is well organized. The contribution relative to previous EEG+ET reading datasets is clearly explained.
Significance and originality
The dataset contains EEG+ET data from 44 participants, making it especially valuable, as the number exceeds that of both previous ZuCo datasets.
The new dataset is in an important field, thus it is both significant and innovative.

Weaknesses

Soundness:
•	Human self-reporting naturally includes a lot of noise. I suspect there were many cases where slight MW occurred while the subjects were unsure of it and thus did not report it. Is it possible, for example, to ask the subjects to report after a paragraph whether the reading was ‘clear’ or ‘unsure’ and then filter out the ‘unsure’ parts?
•	I didn’t understand how MW onset was inferred. It’s possible for the mind to start wandering without awareness, making it difficult to force participants to choose a specific onset word.
•	Why was MW not recorded if it occurred twice on a page?
•	In section 4.2, the split is 80/10/10. Is this done at the article level? I.e., 4 articles in the training, and the fifth is split between val and test.
•	Performance of the baselines across the seven models shows close performance between single modality (ET) and ET+EEG. The best performance on AUROC is 0.609, which is only 0.02 above its single-modality counterpart.
•	This could indicate that a single modality is just as meaningful. Conducting multiple evaluations (possibly with different hyperparameters as well) with a CI would yield greater confidence.
Presentation
•	It’s not clear to me what the structure of the final dataset is. Appendix A.3 clearly shows the features, but what about the time dimension? Is the input time-series? If so, is it ~1 Hour on average for each subject, or is it split up by articles? This might not matter much if the goal is to train classical binary classifiers on MW, but for training larger deep NNs, the structure of the data and how much pre-processing was done are important.
•	Small note: In A.3, can we include a table listing all the features (besides the 64 EEG channels) with a brief description of each?
•	Why do you call a 50-hour dataset "large-scale”? Modern large-scale NLP datasets (not related to neuroscience) are over 100 GB.
•	I didn’t understand the following:
“In a paper currently under review, we showed that incorporating MW onset information significantly improves the performance of linear classifiers trained to detect MW from ET features.”
•	What does MW onset information exactly refer to? Is this related to Appendix B.3?
•	Can you provide more details on the steps taken to turn the Wikipedia articles into the readings for the subjects?
•	Could you elaborate numerically on: “allowing multiple fixations per word would substantially inflate the number of fixation events”
•	The discussion states the best AUROC is 0.601, while Table 3 shows 0.609.

Significance
•	Wikipedia articles, though widely used, still provide limited text variety despite using different articles.
•	While 50 hours is a good amount for a cognitive neuroscience dataset, I believe that highly complex sources of information, such as EEG, require significantly more data to produce high-performance ML models.
Originality
•	It is difficult to justify the reliability of the new addition of MW onset and offset due to the points mentioned above.

---

> ### Author Rebuttal · Authors · 2026-03-28
>
> We thank the reviewer for the constructive feedback and for recognizing the significance and originality of the ROAMM dataset. We address the specific questions below. Due to space constraints, some shared points are addressed in responses to other reviewers. We appreciate your understanding.
>
> ## Paragraph Thought Probes ##
> While paragraph-based thought probes can reduce noise, they significantly disrupt the natural reading flow, which is central to our naturalistic dataset. Moreover, paragraph-level probes are often unable to capture the precise timing of short-duration MW episodes. Relying on thought probes also diminishes the quantity of labeled data: MW episodes may be shorter because they are interrupted, and MW episodes where a thought probe did not occur are not captured.
>
> ## Ambiguous MW Onset ##
> We acknowledge that MW often begins without immediate awareness. In our study, we defined MW as a state that disrupts information processing and comprehension. While this diverges from some MW definitions, we prioritized maintaining a strong connection to reading comprehension. Under this definition, participants can more reliably identify where they need to re-read to recover missed information. We also included practice sessions to ensure participants were comfortable with retrospective reporting.
>
> ## Single MW Report ##
> We limited MW reports to one per page because self-reporting and subsequent re-reading alter the cognitive state. To avoid task-induced artifacts becoming confounding factors in naturalistic reading, we included only the first "clean" instance per page. This conservative design prioritizes label quality over quantity.
>
> ## Data Split ##
> To prevent data leakage, we performed the 80/10/10 split at the unique sentence level, ensuring that sentences in the test set are completely unseen during training. This follows the rigorous approach of Wang & Ji (2024).
>
> ## Multimodal Utility ##
> Please refer to our **"Multimodal Utility"** response to Reviewer **7pkg** for a detailed discussion.
>
> ## Data Structure ##
> The dataset is provided as synchronized raw time series (EEG and ET) with corresponding event markers for fixations and MW labels. We will include an appendix table describing all non-EEG features. It's about one hour for each subject. Additionally, we will release a word-level (fixation-based) EEG feature dataset to facilitate direct use by ML researchers. All raw, unprocessed files will also be made available upon request.
>
> ## Large-Scale Claim ##
> We agree that 50 hours may be considered modest. However, ROAMM would be considered large within the context of multimodal datasets that combine synchronized EEG and eye-tracking with time-resolved cognitive labels. Furthermore, our dataset is collected in a uniquely naturalistic setting, in contrast to more constrained conditions typical of many laboratory-based studies. It is also worth noting that the original recordings were collected at high sampling rates (EEG at 2048 Hz and eye-tracking at 1000 Hz) and later downsampled to 256 Hz for practical use and distribution. If we were to retain the higher sampling rate at 1000 Hz, ROAMM would amount to approximately 150 GB of .pkl files, further reflecting the richness and granularity of the collected data..
>
> We also agree that 50 hours of EEG data may not support training very large or foundation models. Nevertheless, the richness of fine-grained cognitive labels enables effective feature engineering and supports supervised or semi-supervised approaches. Additionally, the dataset can serve as a valuable resource when combined with other EEG+ET datasets for larger-scale training. To avoid potential misunderstanding, we are happy to revise our terminology and avoid the term "large-scale" where it may be misleading.
>
> ## MW Onset Utility ##
> The referenced paper (to be cited after deanonymization) shows that using reported MW onset to define feature extraction windows yields better performance than arbitrary windows (e.g., fixed 2- or 5-second intervals). This is further supported by App B.3, where we observe a decrease in fixation rate, consistent with prior findings, emerging around the reported onset.
>
> ## Wikipedia Processing ##
> Our goal was to select reading material that balances naturalistic engagement with a high likelihood of eliciting MW episodes. We selected five Wikipedia articles spanning diverse topics, manually removing images and specialized jargon to minimize confounds. Each article was standardized to 10 pages to ensure consistency across sessions. Additional details on rendering and display are provided in Section 3.2, and we are happy to provide further clarification if needed.
>
> ## Multiple Fixations ##
> Please see our response to Reviewer **7pkg**, **Word-Level EEG Feature Aggregation**.
>
> ## AUROC Typo ##
> Thank you for catching this typo. We will correct it in the final version.

---

> > ### Author Rebuttal · Reviewer_Pqan · 2026-04-04
> >
> > The authors have done a good job addressing many of my concerns. However, I still have questions about ‘repeated experiments on Table 3’ and the AUROC being close to random chance.

---

> > > ### Author Response · Authors · 2026-04-07
> > >
> > > Thank you for noting that our rebuttal addressed many of your concerns. We address your remaining questions below.
> > >
> > > ## Repeated Experiments on Table 3
> > > Table 3 presents MW classification performance under LOSO evaluation. We interpreted the request for repeated experiments as a question regarding the repeatability and statistical reliability of our MW classification results. We apologize if this interpretation differs from the reviewer’s intent.
> > >
> > > This is a very valid point, and concerns about repeatability and statistical significance were also raised by another reviewer. To address this, we conducted 1,000 iterations of subject-level bootstrapping while maintaining LOSO splits to estimate the distribution of performance metrics. We then calculated p-values by comparing these distributions against a 0.5 chance baseline for our balanced dataset. After applying FDR correction across all models and modalities for each performance metric, the results remained highly significant particularly for Accuracy, Precision, and AUROC.
> > >
> > > The table below summarizes these FDR-corrected p-values. We will also add significance markers (*) to Table 3 in the main paper and include the full statistical significance analysis in the revised manuscript.
> > >
> > > | Model | Feature Set | Accuracy | AUROC | F1 | Precision |
> > > | :--- | :--- | :--- | :--- | :--- | :--- |
> > > | **Gradient Boosting** | EEG | <0.001 | <0.001 | 0.219 | <0.001 |
> > > | | EEG + Eye | <0.001 | <0.001 | 0.093 | <0.001 |
> > > | | Eye | <0.001 | <0.001 | 0.021 | <0.001 |
> > > | **KNN** | EEG | 0.030 | 0.030 | 0.054 | 0.030 |
> > > | | EEG + Eye | 0.007 | 0.007 | 0.028 | 0.007 |
> > > | | Eye | <0.001 | <0.001 | 0.021 | <0.001 |
> > > | **Linear SVC** | EEG | <0.001 | <0.001 | 0.137 | <0.001 |
> > > | | EEG + Eye | <0.001 | <0.001 | 0.065 | <0.001 |
> > > | | Eye | <0.001 | <0.001 | 0.068 | <0.001 |
> > > | **LogReg** | EEG | <0.001 | <0.001 | 0.137 | <0.001 |
> > > | | EEG + Eye | <0.001 | <0.001 | 0.065 | <0.001 |
> > > | | Eye | <0.001 | <0.001 | 0.089 | <0.001 |
> > > | **MLP** | EEG | <0.001 | <0.001 | 0.132 | <0.001 |
> > > | | EEG + Eye | <0.001 | <0.001 | 0.068 | <0.001 |
> > > | | Eye | <0.001 | <0.001 | 0.028 | <0.001 |
> > > | **Random Forest** | EEG | <0.001 | <0.001 | 0.445 | <0.001 |
> > > | | EEG + Eye | <0.001 | <0.001 | 0.222 | <0.001 |
> > > | | Eye | <0.001 | <0.001 | 0.021 | <0.001 |
> > >
> > > ## AUROC Being Close to Random Chance
> > > We acknowledge that an AUROC of 0.609 is modest. However, this reflects a conservative and realistic evaluation, and better represents the performance one should expect when applying the model to unseen subjects.
> > >
> > > To provide additional context, we also trained within-subject MW classifiers, which achieved a highest AUROC of 0.92 using the hybrid modality. Across participants, the mean AUROC was 0.72 (range: 0.57 - 0.92), indicating substantial variability across individuals: https://anonymous.4open.science/r/roamm_ml-CB37/individual_MW_classifier_perf.png
> > >
> > > Although within-subject performance is higher, we intentionally report LOSO results to evaluate true cross-subject generalization rather than subject-specific decoding. Studies that do not adopt LOSO may report inflated performance due to temporal leakage or reliance on subject-specific patterns. We therefore use LOSO as a more rigorous and realistic evaluation setting. **Importantly, despite the modest AUROC values, our results remain consistently above chance and statistically significant, as demonstrated by the bootstrap analysis described above.**
> > >
> > > In addition, our cross-subject performance is consistent with prior work:
> > > - Jin et al. (2019): LOSO AUC ≈ 0.60
> > > - Dong et al. (2021): LOSO AUC = 0.613 vs. within-subject AUC = 0.715
> > > - Tang et al. (2025): LOSO AUC = 0.56 vs. within-subject AUC = 0.68
> > >
> > > These results suggest that the observed performance is in line with the inherent difficulty of cross-subject MW detection, rather than a limitation specific to our dataset.
> > >
> > > Finally, we note that the primary goal of Table 3 is not to achieve SOTA MW detection performance, but rather to **establish a realistic benchmark and demonstrate that ROAMM enables cross-subject MW detection under naturalistic reading conditions.** We hope this dataset will encourage the development of more advanced models for this challenging task and ultimately support applications in real-world settings.
> > >
> > > **Hope this address your concerns. We really appreciate your time and efforts in reviewing our paper!**
> > >
> > > Jin, C. Y., Borst, J. P., and van Vugt, M. K. Predict- ing task-general mind-wandering with eeg. Cognitive, Affective, amp; Behavioral Neuroscience, 19(4):1059– 1073, March 2019.
> > >
> > > Dong, H. W., Mills, C., Knight, R. T., and Kam, J. W. Y. De- tection of mind wandering using eeg: Within and across individuals. PLOS ONE, 16(5):e0251490, May 2021.
> > >
> > > Tang, S., Jiang, C., and Li, Z. Detecting mind wandering via eeg and facial video features. Behavior Research Methods, 57(11), October 2025

---

### Official Review · Reviewer_cjLH · 2026-03-13

**Soundness:** 2
**Presentation:** 3
**Significance:** 3
**Originality:** 3
**Overall Recommendation:** 2
**Confidence:** 4

**Summary:**

This paper introduces ROAMM, a large-scale multimodal dataset for naturalistic reading that combines synchronized EEG and eye-tracking recordings with page-level comprehension scores, eye-event annotations, and time-resolved word-level mind-wandering (MW) labels. The authors benchmark MW detection under leave-one-subject-out evaluation using feature-based supervised classifiers and further test an existing EEG-to-text decoding pipeline under different attention conditions. The authors show that multimodal EEG+ET features improve MW detection over single modalities and that EEG-to-text decoding performance degrades on MW-labeled segments.

**Compliance With Llm Reviewing Policy:**

Affirmed.

**Key Questions For Authors:**

- How sensitive are the MW-detection results to the exact definition of MW offset as 2 seconds before the report button press?
- Did the authors examine whether page topic, article order, or reading fatigue systematically influenced MW frequency or classifier performance?
- Section 3.1: "United State" -> "United States"

**Limitations:**

- The best AUROC for MW detection is 0.609, which is above chance but still relatively modest. This may be acceptable for a difficult task, but the paper should more explicitly frame what constitutes a meaningful baseline.
- The key novelty depends heavily on the validity of self-reported onset/offset words. While the paper provides some supporting evidence, the reliability of these annotations remains a major concern.
- The decoding experiment mainly demonstrates performance degradation with MW, but does not yet establish whether attention-aware modeling can recover performance.

**Strengths And Weaknesses:**

- ROAMM appears to be the first public dataset combining synchronized EEG, eye tracking, and word-level MW onset/offset labels during naturalistic multi-page reading. This is a meaningful contribution.
- The paper makes a strong case that attention fluctuations are a major confound for neural decoding, rather than mere nuisance noise.

---

> ### Author Rebuttal · Authors · 2026-03-28
>
> We thank the reviewer for the constructive feedback and for recognizing ROAMM as a "meaningful contribution" and a "first-of-its-kind" public EEG-ET dataset for naturalistic reading. We address the specific questions below. Due to space constraints, some shared points are addressed in responses to other reviewers. We appreciate your understanding.
>
> ## 2-Sec Offset
> The 2-second removal prior to the report button press is a standard convention to mitigate the influence of "report-related" artifacts. This window is typically contaminated by vertical eye movements and motor-preparatory muscle activity (e.g., mu rhythm) related to the physical button press.
>
> Including these 2 seconds would likely inflate classifier performance due to these non-neural artifacts rather than genuine MW-related cognitive states. By excluding them, we ensure our benchmarks reflect true physiological markers of MW. We will clarify this rationale and cite similar convention (Faber et al., 2017) in the final version.
>
> ## Influence of Page Topic, Article Order, or Reading Fatigue
> This is an excellent question. We tested whether MW frequency was influenced by **page position** (within-story fatigue), **run/trial number** (across-experiment fatigue), and **story** (topic effects) using a logistic mixed-effects model with subject as a random intercept. We found:
>
> - **Run number:** not significant (p = .096)
> - **Page × Run interaction:** not significant (p = .560)
> - **Page number:** marginal trend (p = .060), suggesting at most weak within-story drift
> - **Story effects:** only one story (*Pluto*) showing reduced MW frequency
>
> Overall, these results suggest that MW frequency was **not systematically driven by fatigue or topic differences**. We have not yet examined whether they influence MW classification performance. We are happy to include this additional analysis in the revision if needed.
>
> ### Mixed-Effects Model Results
> | Predictor | β | SE | z | p |
> |-----------|---|----|---|---|
> | page | 0.031 | 0.017 | 1.88 | 0.060 |
> | run | -0.058 | 0.035 | -1.67 | 0.096 |
> | page × run | -0.007 | 0.012 | -0.58 | 0.560 |
> | story: Pluto | -0.478 | 0.152 | -3.14 | 0.0017 |
> | story: Prisoners Dilemma | -0.029 | 0.152 | -0.19 | 0.849 |
> | story: Serena Williams | -0.191 | 0.151 | -1.27 | 0.206 |
> | story: Voynich Manuscript | 0.001 | 0.152 | 0.00 | 0.996 |
>
> ## 0.609 AUROC and Meaningful Baselines
>
> Although some studies report MW detection performance above 0.9, an AUROC of 0.609 is consistent with many prior findings using LOSO. Several factors contribute to this:
>
> **Intrinsic Complexity of MW**
> MW is a spontaneous, internal, and often unconscious cognitive state. Detecting transitions using current paradigms inevitably introduces noisy ground truth. Furthermore, unlike highly controlled tasks (e.g., SART), ROAMM involves naturalistic multi-page reading, where cognitive states are more complex and neural signals are inherently noisier.
>
> **Evaluation Rigor (LOSO)**
> As discussed with Reviewer **fRKr**, we intentionally used LOSO because it provides the most conservative and realistic evaluation for cross-subject deployment. We also reported within-subject performance in our response to Reviewer **7pkg** (under **Inter-Subject Generalization**), showing that performance improvements are consistent with prior work.
>
> We will add literature comparisons for cross-subject MW detection and clarify what constitutes meaningful performance in this challenging setting.
>
> ## Reliability of MW Labels
>
> Capturing the precise MW time is a known challenge in the field. Traditional self-report or thought-probe paradigms often inaccurately assume MW begins at trial onset or 2 or 5s before. Our introspective paradigm enables moment-to-moment annotations. To validate these self-reports, we examined objective physiological changes surrounding reported MW onset (App B.3). Specifically, our temporal dynamics analysis shows a significant drop in fixation rate aligned with the reported MW onset. This alignment between subjective reports and objective eye-tracking metrics provides strong evidence for the validity of our labels.
>
> ## Attention-Aware Modeling
>
> The reviewer raises an excellent point. We lack the data quantity to train an EEG-to-text decoder that can condition its parameters on MW state, so our aim is to inform readers what level of performance is lost by including MW episodes in the decoding process. In this study, we excluded MW-labeled segments, a naïve but effective form of attention-aware filtering, and showed in Table 4 that decoding performance improves on non-MW segments.
>
> We agree that end-to-end attention-aware decoding models that weight or gate inputs based on predicted attention is an important next step. We will explicitly frame this as a future research direction.
>
> Faber, M., Bixler, R., and D’Mello, S. K. An auto-mated behavioral measure of mind wandering during computerized reading. Behavior Research Methods, 50 (1):134–150, February 2017.

---

> > ### Author Rebuttal · Reviewer_cjLH · 2026-04-03
> >
> > The rebuttal is helpful but does not resolve my main concerns. The paper’s core novelty still relies heavily on the reliability of retrospectively self-reported MW onset/offset labels, and the added evidence remains indirect. The benchmark strength and EEG-to-text evaluation also remain limited. I therefore keep my original score.

---

> > > ### Author Response · Authors · 2026-04-07
> > >
> > > Thank you for noting that the rebuttal was helpful in resolving some concerns. Here we would like to further elaborate on the remaining concerns.
> > >
> > > ## ROAMM Core Novelties
> > > We believe ROAMM introduces several key novelties beyond existing datasets. First, there are currently only two widely used public datasets that provide simultaneous EEG and ET during English reading. These datasets, ZuCo and ZuCo 2.0, have been widely used in both the BCI and ML communities, especially for EEG-to-text decoding. **ROAMM contributes by providing another high-quality simultaneous EEG and ET dataset**, expanding opportunities for model development and benchmarking.
> > >
> > > **Second, ROAMM adopts a more naturalistic reading paradigm.** ROAMM allows free reading of multi-page text, which more closely reflects real-world reading behavior. Although increased ecological validity introduces additional challenges, we addressed them through careful preprocessing, standard artifact removal, and extensive manual quality assurance (e.g., removal of 2 seconds prior to butter press) to ensure high data quality.
> > >
> > > **Third, ROAMM provides MW annotations during naturalistic reading.** Capturing MW is inherently difficult, and current gold-standard approaches are self-reports or thought probes. To our knowledge, no publicly available dataset currently combines simultaneous EEG and ET with MW annotations during naturalistic reading. ROAMM therefore represents the first public dataset in this setting.
> > >
> > > **Finally, ROAMM introduces span-level MW annotations using a retrospective reporting paradigm motivated by real-world reading behavior.** When readers realize they were not paying attention, they naturally re-read previously missed content to regain comprehension. By allowing participants to retrospectively mark these spans, we obtain better estimates of MW onset and offset. Prior literature has shown reduced fixation rate during ambiguous MW periods, and our temporal analysis (Appendix B.3, Figure 7) shows that fixation rate decrease starts around the self-reported MW onset. This alignment between subjective reports and objective eye-tracking measures provides supporting evidence for the validity of our labels.
> > >
> > > ## MW Classification
> > > We acknowledge that an AUROC of 0.609 is modest. However, this reflects a conservative and realistic evaluation, and better represents the performance one should expect when applying the model to unseen subjects.
> > >
> > > To provide additional context, we also trained within-subject MW classifiers, which achieved a highest AUROC of 0.92 using the hybrid modality. Across participants, the mean AUROC was 0.72 (range: 0.57 - 0.92), indicating substantial variability across individuals: https://anonymous.4open.science/r/roamm_ml-CB37/individual_MW_classifier_perf.png
> > >
> > > Although within-subject performance is higher, we intentionally report LOSO results to evaluate true cross-subject generalization rather than subject-specific decoding. Studies that do not adopt LOSO may report inflated performance due to temporal leakage (also justified by **Reviewer fRKr**) or reliance on subject-specific patterns. We therefore use LOSO as a more rigorous and realistic evaluation setting.
> > >
> > > Importantly, our cross-subject performance is consistent with prior work:
> > >
> > > - Jin et al. (2019): LOSO AUC ≈ 0.60
> > > - Dong et al. (2021): LOSO AUC = 0.613 vs. within-subject AUC = 0.715
> > > - Tang et al. (2025): LOSO AUC = 0.56 vs. within-subject AUC = 0.68
> > >
> > > These results suggest that the observed performance is in line with the inherent difficulty of cross-subject MW detection, rather than a limitation specific to our dataset.
> > >
> > > ## EEG-to-text Evaluation
> > > Getting the highest EEG-to-text decoding performance is not the primary focus of this dataset paper. Instead, we adopt an existing ZuCo-based baseline to demonstrate two key points: (1) decoding is feasible on ROAMM, and (2) decoding performance decreases during MW periods, highlighting the importance of cognitive state in neural decoding.
> > >
> > > Lower decoding performance compared to ZuCo is expected. Naturalistic settings introduce greater behavioral variability and lower signal-to-noise ratio, which can negatively impact decoding performance.
> > >
> > > ## Summary
> > >
> > > Overall, ROAMM contributes:
> > >
> > > - A high-quality simultaneous EEG and ET dataset
> > > - A naturalistic multi-page reading paradigm
> > > - The first public dataset with MW labels in this setting
> > > - Span-level MW annotations
> > > - Rigorous LOSO evaluation
> > >
> > > We believe ROAMM provides substantial value for both the neuroscience and ML communities by enabling research that is difficult to conduct using more constrained laboratory datasets and by supporting the development of models that better reflect real-world reading behavior.
> > >
> > > We sincerely appreciate your time and effort in reviewing our paper and thank you for your thoughtful consideration.

---

### Official Review · Reviewer_fRKr · 2026-03-13

**Soundness:** 4
**Presentation:** 4
**Significance:** 3
**Originality:** 3
**Overall Recommendation:** 5
**Confidence:** 4

**Summary:**

This work introduces a rich EEG + EyeTracking (ET) + MindWandering (MW) annotated reading dataset, with 44 subjects and 50 hours of data. The dataset is validated in two respects: (1) The MindWandering annotations can be decoded with above chance accuracy from EEG+ET signals in a rigorous Leave-One-Subject-Out fashion (2) The text can be decoded from EEG better when not MW

**Compliance With Llm Reviewing Policy:**

Affirmed.

**Final Justification:**

While I agree with Reviewer cjLH that retrospective self-reports can be unreliable, the subjective nature of mind-wandering does force the use of such methods for data collection. Given that the authors conducted rigorous LOSO eval and still found statistically significant effects, I think it's a technically sound dataset that could be nice for future work to analyze effects of MW on decoding performance. I recommend acceptance.

**Key Questions For Authors:**

1. For the MW classification, is the MW-label-decoding statistically significant?

**Limitations:**

Yes

**Strengths And Weaknesses:**

**Soundness**: The work is technically sound and rigorous, barring one question (see below) - appropriate care has been given to the evaluation, in which Leave-One-Subject-Out is crucial. Because MW annotations are temporally correlated (a block of text is labeled MW), and temporal information can be trivially decoded from EEG, any evaluation that splits the data arbitrarily into train/test will achieve extremely high (>90%, and confounded) accuracy.
1. For the MW classification, is the MW-label-decoding statistically significant?

**Presentation**: The work is written well and is clear. However, "word-level MW annotations" is often used in the text, which gives the incorrect impression that MW words are interleaved with non-MW words in the annotations. This should be changed to "span-level" or similar to be more accurate.
Finally, a multitude of EEG-related decoding papers (particularly from ML practitioners less familiar with EEG) in various fields have reported extremely high decoding performance due to the reasons mentioned above. Since this dataset has the same potential to produce such works, I recommend making it very clear in the manuscript why evaluation should be done in via LOSO.

**Significance**: This will be a rich dataset for the members of the EEG community as well as ML practitioners interested in neural decoding, and can support many interesting research questions.

**Originality**: The work introduces a new dataset and is novel in that respect. It also considers attention-mediated text decoding, which has received limited attention.

---

> ### Author Rebuttal · Authors · 2026-03-28
>
> We thank the reviewer for the positive assessment of our dataset and the rigor of our methodology. We address the specific points below.
>
> ## Statistical Significance
>
> To address the question on significance, we conducted 1,000 iterations of subject-level bootstrapping (maintaining LOSO) to estimate metric distributions. We calculated p-values by comparing these distributions against a 0.5 chance baseline for our balanced dataset. After False Discovery Rate (FDR) correction across all models and modalities for each figure of merit, the results remain highly significant (typically *p* < .001), particularly for Accuracy, Precision, and AUROC.
>
> The table below summarizes these FDR-corrected p-values. We will add significance markers (*) to our main results table and include this full significance analysis in the revision.
>
>
> ### FDR Corrected P-Values (Baseline = 0.5)
>
> | Model | Feature Set | Accuracy | AUROC | F1 | Precision |
> | :--- | :--- | :--- | :--- | :--- | :--- |
> | **Gradient Boosting** | EEG | <0.001 | <0.001 | 0.219 | <0.001 |
> | | EEG + Eye | <0.001 | <0.001 | 0.093 | <0.001 |
> | | Eye | <0.001 | <0.001 | 0.021 | <0.001 |
> | **KNN** | EEG | 0.030 | 0.030 | 0.054 | 0.030 |
> | | EEG + Eye | 0.007 | 0.007 | 0.028 | 0.007 |
> | | Eye | <0.001 | <0.001 | 0.021 | <0.001 |
> | **Linear SVC** | EEG | <0.001 | <0.001 | 0.137 | <0.001 |
> | | EEG + Eye | <0.001 | <0.001 | 0.065 | <0.001 |
> | | Eye | <0.001 | <0.001 | 0.068 | <0.001 |
> | **LogReg** | EEG | <0.001 | <0.001 | 0.137 | <0.001 |
> | | EEG + Eye | <0.001 | <0.001 | 0.065 | <0.001 |
> | | Eye | <0.001 | <0.001 | 0.089 | <0.001 |
> | **MLP** | EEG | <0.001 | <0.001 | 0.132 | <0.001 |
> | | EEG + Eye | <0.001 | <0.001 | 0.068 | <0.001 |
> | | Eye | <0.001 | <0.001 | 0.028 | <0.001 |
> | **Random Forest** | EEG | <0.001 | <0.001 | 0.445 | <0.001 |
> | | EEG + Eye | <0.001 | <0.001 | 0.222 | <0.001 |
> | | Eye | <0.001 | <0.001 | 0.021 | <0.001 |
>
>
> ## Word Choice
>
> We agree with the reviewer's suggestion. To avoid any ambiguity regarding the continuity of mind-wandering episodes, we will replace **"word-level"** with **"time-resolved"** (or **"moment-to-moment"**) MW annotations.
>
> ## LOSO Justification
>
> We strongly agree that arbitrary train/test splits lead to confounded results. Beyond mitigating temporal leakage (which can inflate accuracy for short trials), our use of LOSO was specifically intended to prevent the model from exploiting subject-specific neural traits, ensuring the generalizability of our findings.
>
> As suggested, we will make it clear in the manuscript why evaluation should be conducted using LOSO.

---

> > ### Author Rebuttal · Reviewer_fRKr · 2026-04-03
> >
> > My primary concerns about statistical significance and technical soundness have been addressed. My prior score already factored this in, so I'll keep it.
> >
> > However, please note that "time-resolved" or "moment-to-moment" is arguably **worse** than "word-level" MW annotations. Please use span-level annotations since that is precisely how you asked the subjects to annotate the data, the other names are quite misleading.

---

> > > ### Author Response · Authors · 2026-04-03
> > >
> > > Thank you very much for this further clarification! We agree that **span-level** is a more accurate description of how the annotations were collected, since participants explicitly marked the onset and offset of mind-wandering episodes. We will therefore adopt the term **span-level MW annotations** throughout the manuscript to better reflect the task design and avoid potential confusion or misleading interpretations.
> > >
> > > We appreciate this suggestion and thank you again for your careful reading and constructive feedback!

---

### Decision · Program_Chairs · 2026-04-30

**Decision:**

Accept (regular)

**Comment:**

This work contributes a dataset of EEG and eye-tracking recorded during naturalistic multi-page reading from 44 participants. The dataset also contains annotations for comprehension scores and self-reported mind-wandering events. The work contains two main evaluations: one of classifying mind wandering events, based on the self-reported annotations, and another of text decoding from the EEG signals. The strength of the work is in providing the new dataset, which would be the first of its kind in magnitude for detecting mind wandering from EEG. The reviewers mostly agree that this is a good resource for cognitive science. The manuscript can be strengthened by including the additional significance results provided during the rebuttal. One outstanding concern raised by the most negative reviewer is the reliability of the self-reported mind wandering events. The authors acknowledge the difficulty of verifying these self-reported measures but have done a reasonable validation effort. Overall, the richness of the new dataset and experimental setting provided by this work make this paper a good contribution.